# Semi-Supervised Behavior Labeling Using Multimodal Data during Virtual Teamwork-Based Collaborative Activities

**DOI:** 10.3390/s23073524

**Published:** 2023-03-28

**Authors:** Abigale Plunk, Ashwaq Zaini Amat, Mahrukh Tauseef, Richard Alan Peters, Nilanjan Sarkar

**Affiliations:** 1Department of Electrical and Computer Engineering, Vanderbilt University, Nashville, TN 37240, USA; 2Department of Mechanical Engineering, Vanderbilt University, Nashville, TN 37240, USA

**Keywords:** human-behavior sensing, emotion recognition, human–machine interaction, automated labeling, semi-supervised machine learning

## Abstract

Adaptive human–computer systems require the recognition of human behavior states to provide real-time feedback to scaffold skill learning. These systems are being researched extensively for intervention and training in individuals with autism spectrum disorder (ASD). Autistic individuals are prone to social communication and behavioral differences that contribute to their high rate of unemployment. Teamwork training, which is beneficial for all people, can be a pivotal step in securing employment for these individuals. To broaden the reach of the training, virtual reality is a good option. However, adaptive virtual reality systems require real-time detection of behavior. Manual labeling of data is time-consuming and resource-intensive, making automated data annotation essential. In this paper, we propose a semi-supervised machine learning method to supplement manual data labeling of multimodal data in a collaborative virtual environment (CVE) used to train teamwork skills. With as little as 2.5% of the data manually labeled, the proposed semi-supervised learning model predicted labels for the remaining unlabeled data with an average accuracy of 81.3%, validating the use of semi-supervised learning to predict human behavior.

## 1. Introduction

Human–machine interaction is a broad field of research that investigates the interaction between human interfaces and dynamic systems [1]. In applications of human–machine interaction, such as virtual reality and computer-based training and interventions, the machine must be able to measure and react to human emotions to enhance real-time feedback and allow for more adaptive systems. Current solutions require manual data labeling by human observers to train machine learning models that can predict human engagement behavior. This process is time-consuming and limits accessibility across low-resource settings, making robust automated labeling essential to the success of such systems [2]. For example, Liu et al. developed two computer-based cognitive tasks that invoke three affective states for intervention in children with autism spectrum disorder (ASD). We will interchangeably use the terms ‘autistic individuals’ and ‘individuals with ASD’ to respect both views on identity-first and person-first language [3,4]. These tasks required manual labeling by therapists to develop a supervised affective model [5]. An additional study developed an immersive virtual reality system to train emotional skills in students with ASD. One goal of the system is to update the social situations according to the student’s emotions. To implement this real-time state change, an evaluator is required to recognize the emotion of the child and guide them in the environment [6]. On the other end of the spectrum, unsupervised machine learning methods, which do not require labeled data, may work on occasion in these interventions, but they are not as reliable. O’Hara et al. used unsupervised learning to classify human expressions, gestures, and actions. While they did achieve high performance with different unsupervised methods, the generalizability of the models varied because they were sensitive to the subject identity [7]. Semi-supervised machine learning is an alternative that does not require as much manual labeling as supervised learning, but is not as sensitive as unsupervised learning.

Semi-supervised machine learning is a technique used to predict the labels associated with partially labeled data. It is a mixture of unsupervised learning, which trains a model using unlabeled data, and supervised learning, which uses a fully labeled dataset. There are two branches of semi-supervised learning: inductive and transductive. Inductive methods aim to find a classification model for all of the data points, while transductive methods are only concerned with assigning labels to the unlabeled data points [8]. For our application, we need a trained classification model so we can utilize it on new data; therefore, we will implement an inductive model. One common methodology for semi-supervised learning is self-training, which is an inductive wrapper method. The algorithm iteratively trains a supervised learning model to create pseudolabels which are predictions with high confidence [9]. Self-training can be used to supplement manual data annotation in labeling human behavior, which is necessary to enable more adaptive systems. 

Semi-supervised learning is used in literature in a wide variety of applications to supplement manual data labeling. One study used semi-supervised learning with multimodal data to classify the manic states of patients with bipolar disorder [10]. Another study investigated supplementing manually labeled driver distraction data with unlabeled data. Using this mix of labeled and unlabeled data, they developed a semi-supervised classification model that predicts driver distraction with higher accuracy than completely supervised models. This is because the semi-supervised models were less prone to overfitting, allowing them to generalize better on unseen data [11]. Finally, a study used self-training for multimodal emotion recognition, which showed significantly better results than using semi-supervised learning with unimodal data [12]. While this machine learning methodology can be applied to many applications, this paper evaluates the use of semi-supervised learning to supplement manual data annotation of human behavior in a teamwork training task for young adults with ASD.

ASD is a neurodevelopmental disorder influenced by environmental and genetic factors that is classified by challenges in social communication and interaction [13]. These challenges result in difficulties when working in teams, leading to obstacles in securing and retaining employment. Approximately 4 million of the 5.4 million adults with ASD in the United States are either unemployed or under-employed, relative to their abilities [14]. Studies have shown that unemployment can lead to increased stress, depression, and anxiety, as well as reduced self-esteem [15,16]. Therefore, it is essential to address deficits in social interactions as they cast a shadow on outstanding qualities, such as precise technical abilities, high tolerance for repetitive tasks, and reliability that autistic individuals can bring to companies [17,18]. Recently, virtual reality and computer-based training and interventions for individuals with ASD have been heavily researched because of their replicability and modifiability. Collaborative virtual environments (CVEs) capitalize on the strengths of virtual training and can be used to teach teamwork skills to autistic individuals. Teamwork skills are associated with improved productivity and workplace performance [19]. They are among the core skills sought by employers and can influence hiring decisions [20]. Therefore, supporting autistic adults to acquire work-relevant teamwork skills may contribute to not only job acquisition, but also improve workplace social communication skills [21], problem-solving [22], and self-confidence [23]. Previous studies have shown the efficacy in teamwork training for adolescents with ASD. For example, community-based vocational programs can be instrumental in teaching technical and soft skills [24]. However, these programs are uncommon and expensive, resulting in an inability to receive the resources necessary to obtain meaningful employment [23]. An alternative to traditional teamwork training is virtual reality (VR)-based training. VR has been shown to be an effective tool for training social, emotional, and daily living skills [25,26]. One study utilized a virtual environment course to teach college students creative thinking. The virtual environment improved the student’s empathy and problem-solving abilities [27]. An additional study utilized virtual reality for social cue detection which enabled the participants to practice rapport [28]. Building on these studies, our previous work developed CVE-based teamwork training tasks. However, to effectively encourage teamwork skills, the CVE must provide real-time prompts based on the user’s current behavior state (engaged, struggling, or waiting) [29]. In our previous workshop paper, we introduced automated behavior labeling during collaborative tasks using an unsupervised K-means clustering model to predict participant behavior which showed promising results on the preliminary data [30]. However, due to the rigidity of unsupervised learning, additional methods to supplement manual data labeling are necessary. This work investigates the more advanced and adaptive method of semi-supervised self-training.

Current research in the field of human–machine interaction relies on manual data labels or unsupervised learning to implement adaptive systems. Our work seeks to address the limitations of these two methods and offer an alternate solution by developing a semi-supervised self-training affective model. The contributions of this paper are as follows:In this paper, we expand our novel multimodal dataset of autistic and neurotypical individuals working together to complete a collaborative task in virtual reality.Using this dataset, we developed a semi-supervised self-training affective model. In doing so, we determined the percentage of labeled data needed for consistent high-accuracy results between our model and ground-truth labels.We compared the performance of our semi-supervised model to both a supervised and unsupervised model to prove the effectiveness of this model. Our semi-supervised model outperformed the unsupervised model and performed comparably to the supervised model. The semi-supervised model is an improvement over the supervised model when considering the trade-off between performance accuracy and manual data labeling.

The following section summarizes the experimental design and data collection. Section 3 describes the methods used for semi-supervised self-training and establishes metrics for performance evaluation. Section 4 analyzes the results and discusses their implications. Finally, the paper is concluded with a summary of the contributions of this work as well as potential future research directions.

## 2. System Design

The following subsections describe the system design and data collection protocol for the collaborative virtual environment. 

### 2.1. Collaborative Tasks

Multimodal data were collected across three collaborative tasks designed and developed in Unity3D that simulate a work environment [31]. The three tasks include a furniture assembly task, a PC assembly task, and a fulfillment center task as shown in Figure 1. In the furniture and PC assembly tasks, two participants worked together to assemble furniture and a PC, respectively. In the fulfillment center task, both participants were tasked with transporting a pallet in the warehouse from the shelf to a loading area using a forklift. We employed a participatory design process where we engaged with stakeholders and end-users from various backgrounds to design meaningful collaborative tasks. We worked with industry representatives from two companies, a certified behavioral interventionist, two career counselors from two vocational rehabilitation centers, and three autistic adults. Stakeholders were involved in both the design and development stages of the collaborative tasks. These tasks were designed to incorporate numerous collaboration principles across all tasks [32]. The participants needed to exchange information, take turns, and coordinate their movements together to achieve the goals in each task. Table 1 describes each collaborative task and their various aspects of collaboration. The design, development, and system architecture of the three collaborative tasks used in this work were presented in detail in our previous work [29].

### 2.2. Data Collection and Processing

Data were collected in a system validation study with six pairs of participants (twelve total participants) working in CVE-based teamwork tasks described in Section 2.1. The pairs consisted one individual with ASD and one neurotypical individual that were matched based on age and gender. Table 2 lists participant information. The study required the participants to work in pairs to complete three collaborative tasks designed to encourage teamwork and collaboration skills. To begin the study, participants were informed about the experiment before completing consent forms. The participants were then directed to separate rooms to have their eye trackers calibrated before logging on to the shared virtual environment. This study was approved by the Institutional Review Board at Vanderbilt University.

Figure 2 shows the collaborative virtual environment experimental setup. The participants joined a shared virtual environment in separate rooms. Each participant had three modalities for data capture: a headset, a controller, and an eye tracker. The game controller was task dependent. A haptic controller was used in the furniture assembly task, a mouse and keyboard were used in the PC assembly task, and a gamepad was used in the fulfillment center task. Based on the methods for feature extraction in [18], binary features were extracted from these three modalities to capture aspects of teamwork. From the headset, speech data were extracted to capture the interaction between the participants. Four features were extracted from the game controller to determine if the participants were actively moving toward the goal or away from the goal. Finally, two features were extracted from the eye tracker to capture where each participant was focused. The features extracted from each modality and their descriptions are listed in Table 3. 

### 2.3. Manual Data Labeling

After extracting features, the data were manually annotated to provide ground-truth labels to validate the proposed method for semi-automated labeling. Three classes of behavior were chosen to encapsulate various aspects of teamwork in the CVE in consultation with behavior professionals. The first two behaviors chosen were engaged and struggling. Engaged captures active collaboration between the participants and involvement with the collaborative task. Struggling occurs when the participant is either not interacting with the system, not advancing toward the goal, or disengaging with their partner. However, while participating in collaborative tasks, there were times when the participants were neither engaged nor struggling. Therefore, an additional behavior, waiting, was added. Waiting represents taking turns during teamwork. This behavior occurs when a participant is not actively working toward the goal, but their partner is. Under the guidance of a behavior professional, two individuals that were familiar with the experimental protocol labeled the six sessions of data independently and compared labels to ensure consistency. The labels were determined using a flow chart of coding rules shown in Figure 3 in correspondence with session videos. This flow chart did not always provide the annotators with a clear class choice. In this case, they decided on the class based on the video and their perception. After labeling all six sessions separately, the labels were compared. The two annotators reached a 98% agreement. The disagreements were settled through discussion. The class distributions of the three behaviors for the six labeled sessions were as follows: engaged—15.37%, waiting—50.27%, struggling—34.36%. The following section details the methods that were taken to develop a semi-supervised model that supplements manual data labeling.

## 3. Methods

The following subsections explain the semi-supervised algorithm used to automate behavior labeling in a collaborative virtual environment and the performance metrics that will be used to evaluate the results. The semi-supervised algorithm was written in MATLAB [33].

### 3.1. Semi-Supervised Self-Training Algorithm

Figure 4 summarizes the semi-supervised self-training algorithm used to supplement manually labeled data in a sparsely labeled dataset. However, our dataset was fully labeled to validate this algorithm. When training the model, a minimal set of labeled data was selected at random, and the remaining labels were withheld for comparison with the model’s predictions. The amount of labeled data supplied to the model is discussed in the following section. 

The algorithm begins by training a supervised machine learning model on the minimal labeled subset of the multimodal data. The supervised model chosen was a support vector machine (SVM) with a Gaussian kernel. The SVM algorithm aims to find the hyperplane that best separates the classes of data points. This is achieved through optimizing the hyperplane that minimizes the soft margin between classes [34,35]. We chose this model because SVMs have a low computational complexity while providing high accuracy results in many applications [36]. We utilized MATLABs built in SVM functions for this dual-optimization classification approach [33]. MATLAB automatically optimizes the hyperparameters during training. After training the model on the small subset of labeled data, it is then used to make predictions on the remaining unlabeled data. The model returns both the predictions and the confidence of each prediction. The confidence of the predictions is the posterior probability of the prediction. The posterior probability of inseparable classes is the sigmoid function:(1)$$P\left(s_{j}\right)=\frac{1}{1+exp(As_{j}+B)}$$
where *A* and *B* are the slope and y-intercept of the hyperplane [37]. The confidence of each prediction is checked, and if it is greater than 50%, that prediction is added as a pseudolabel to the labeled dataset. A total of 50% was chosen to ensure that no classes were equally probable (e.g., engaged = 50%, waiting = 50%, struggling = 0%) and to ensure as little ambiguity between classes as possible. A new supervised model is trained using the updated labeled dataset consisting the original ground-truth labels and the new pseudolabels. The process of checking the confidence of the prediction and adding pseudolabels to the labeled set is continued until over half of the data points are in the labeled dataset. Due to the ambiguity of human behavior, there is not always a clear behavior state. Therefore, as the process is iterated, the remaining data are more ambiguous, meaning that fewer predictions reach the threshold confidence and fewer pseudolabels are added to the labeled dataset. After fifty percent of the data points were in the labeled dataset, the confidence of the predictions rarely exceeded the threshold.

After completing this process, the final labeled dataset is compared to the ground-truth labels to determine how well the model performed. The following subsection discusses the performance metrics used to determine model performance.

### 3.2. Performance Metrics

When training the semi-supervised model, the labeled data subset was chosen randomly. Depending on the subset chosen, the class distribution of behavioral states (i.e., engaged, waiting, and struggling) varied. This resulted in a highly variable performance across multiple models that was dependent on the labeled data supplied at the very beginning. Additionally, the performance depended on the percentage of labeled data supplied at the beginning. Therefore, when evaluating performance, both dependencies needed to be considered.

The dataset contained six fully labeled sessions. Four sessions were chosen randomly to develop the training dataset for the machine learning model, and the final two sessions were used as a hold-out test set. The chosen training set contained 2014 data points and the test set contained 1487 data points. Within the training set, the labeled and unlabeled subsets were supplied randomly. A total of 500 trials were completed with different labeled subsets of data supplied at the beginning to take into effect the first dependency. Additionally, four sub-trials were completed to investigate the amount of labeled data needed to develop a reliable model to evaluate the second dependency. These sub-trials supplied the training set with 2.5% labeled data, 5% labeled data, 10% labeled data, and 25% labeled data.

To evaluate the performance of each sub-trial, the accuracy of all 500 trials was considered as well as the average accuracy across the 500 trials. Ideally, most of the trials would result in high accuracy predictions on the test set with few trials resulting in low accuracy predictions. We defined high accuracy to be above 80% and low accuracy to be below 70%. We also hoped to achieve this with as small of a subset of labeled data as possible. The final consideration was the performance of self-training semi-supervised learning versus the performance of our previous work using unsupervised learning and fully supervised methods. The results are presented and discussed in the following section.

## 4. Results and Discussion

To begin, we wanted to determine the amount of labeled data needed for consistent results using our proposed method of semi-supervised self-training. Four sub-trials were completed with varying amounts of labeled data supplied. Each sub-trial consisted 500 trials with randomly supplied ground-truth labels. Figure 5 shows the results across all trials. The test sessions used the model trained on the denoted percentage of labeled data in the training set. No labeled data were supplied from the test sessions. 

As shown in Figure 5, with as little as 2.5% of the ground-truth labels supplied, over 75% of both the training and test trials resulted in models with greater than 80% accuracy. By doubling the supplied labels to 5%, the average accuracy increases by approximately 2.5%. After that, the average accuracy levels off to the maximum possible, and there are few trials with an accuracy of less than 80% accuracy. Therefore, with any random 10% of the total data having ground-truth labels supplied, semi-supervised machine learning can train a model that has maximum predictive accuracy.

In addition to accuracy, it is important to ensure that all the classes are being predicted. Engaged only constitutes around 15% of the data; therefore, it is possible to achieve over 80% accuracy without ever predicting engaged. Figure 6 shows the confusion matrices for both the training and test sessions using predictions made with the semi-supervised model trained on 10% of the training data. As shown, the semi-supervised model occasionally incorrectly distinguishes between waiting and struggling, but otherwise consistently predicts the correct labels for the remaining data. The misclassification between struggling and waiting is due to the nuanced differences between these two classes. In addition to the confusion matrices, we looked at state progression charts for both participants to see when misclassifications occur. Figure 7 consists the state progression charts for one of the training sessions and one of the test sessions. As shown in the figure, the predictions from the semi-supervised model closely follow the trend of the actual behavior changes during the collaborative tasks. The misclassifications, in general, happen when the participant is quickly moving back and forth between two behavior states. The model over-predicts waiting during these scenarios. For our application, this is ideal behavior. The future application of this work is to provide real-time feedback based on the predictions of the semi-supervised model. If the participant is struggling, the system will prompt the users to work together, and if the users are engaged, the system will provide positive reinforcement. When the participants are waiting, the system will not interfere. If the state is ambiguous, the system should not interfere as it may cause confusion. Therefore, the model predicting waiting in uncertain times allows the system to function as intended.

Our results imply that semi-supervised machine learning can be used to supplement manual data labeling of human behavior with consistent high-accuracy results. Based on this, the results can be compared to our previous work using unsupervised K-means clustering to manually label a behavior state in the same CVE workplace. The methods and results using clustering are discussed in detail in our previous work [30]. The unsupervised clustering results presented here deviate slightly from the results in our previous workshop paper because more sessions of data have been collected. Therefore, the previous code was repeated using the same four sessions of data for training and two sessions as a test to mimic the protocol for our semi-supervised model. In addition to comparing our semi-supervised method to our previously developed unsupervised method, we wanted to compare it to supervised learning models as a baseline. Therefore, we tested three different supervised models for comparison. Figure 8a shows the accuracy across three different supervised models. The best model, which is a support vector machine with a Gaussian kernel, is compared to the unsupervised and semi-supervised model in Figure 8b. The semi-supervised accuracy shown is achieved using the model with 10% ground-truth labels from Figure 5. 

It is shown that a completely supervised model outperforms both the unsupervised and semi-supervised models, which is to be expected. However, the self-training test accuracy is comparable to the supervised model, supporting that semi-supervised learning methods can achieve high-accuracy results with limited labeled data. When comparing only unsupervised learning and semi-supervised learning, the training accuracy is higher using unsupervised learning, but the test accuracy is higher for self-training. Clustering is more rigid than self-training, which can lead to overfitting and less generalizability on new data, while self-training is less rigid. Based on this, we expected self-training to generalize better on unseen data than K-means which is demonstrated in these results. There is a trade-off between supplying a subset of ground-truth labels resulting in generalizability to new data and providing no ground-truth labels and having less generalizability. In general, human behavior is highly variable. Two individuals could both be *engaged* in collaborative tasks while still acting differently. Therefore, generalizability is important in our application, making semi-supervised self-training a reliable option for supplementing manual data labeling.

## 5. Conclusions

In this paper, we discussed the use of semi-supervised self-training to complement the manual labeling of human behavior. Although manual labeling is a reliable source for ground-truth labels, it can be time-consuming and resource-straining. As such, there is a need for an alternative method to automatically analyze the interpersonal social behavior of the users in team-based tasks. 

This paper investigated collaborative multimodal data using semi-supervised machine learning to label users’ interpersonal behavior during team-building tasks in a CVE. In doing so, we determined the percentage of ground-truth labels needed to generate a reliable semi-supervised model in our application. With as little as 2.5% of the labels supplied, the average accuracy across all data was 81.8%. However, there were low accuracy models in the 500 random trials, which means that if the labels were supplied at random, this method may not yield a model that generalizes well on new data. By increasing the number of labels to 10%, semi-supervised machine learning resulted in consistent high accuracy results across all 500 trials with an average test accuracy of 84.5%. Finally, we validated the use of semi-supervised self-training against hand-labeled data and compared the results to our previous method using K-means clustering and a fully supervised SVM model. Semi-supervised learning enabled accuracy within 1.6% of a fully supervised model on the hold-out test set and improved upon the unsupervised model by 4.1%. Due to its generalizability on new data and a reduced need for tedious manual labeling, it is the best choice model for predicting behavior states in a CVE to eventually provide real-time prompts that encourage teamwork. 

The ability to automate behavior labeling in human–machine systems is critical for enhancing real-time feedback and making adaptive systems. With the methodology described in this work, we were able to develop a reliable affective model with as little as 2.5% of the data manually labeled, reducing the workload and human error drastically. In doing so, we can now implement a closed-loop feedback mechanism to provide real-time feedback that encourages teamwork based on participant engagement. Our hope is that this mechanism will enhance skill learning and provide an accessible tool for young adults with autism to practice collaborative skills. 

While the results discussed above show promise, it is essential to highlight the limitations of the study and important targets for future research. First, our sample size was relatively small. Recruiting more participants would allow us to further validate the robustness of our model. Next, the model utilized does not consider temporal information that human behavior can depend on. Additionally, our model is deterministic, meaning the output is always the same for the same input. However, this is not always true for human behavior which could contribute to the misclassification of the waiting and struggling states. Based on the results described in this paper, future work will implement a closed-loop feedback mechanism that provides real-time feedback based on the participant’s behavior as determined using semi-supervised machine learning.

## Figures and Tables

**Figure 1 sensors-23-03524-f001:**
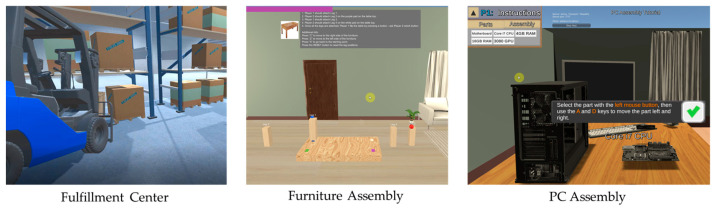
Collaborative tasks used for data collection.

**Figure 2 sensors-23-03524-f002:**
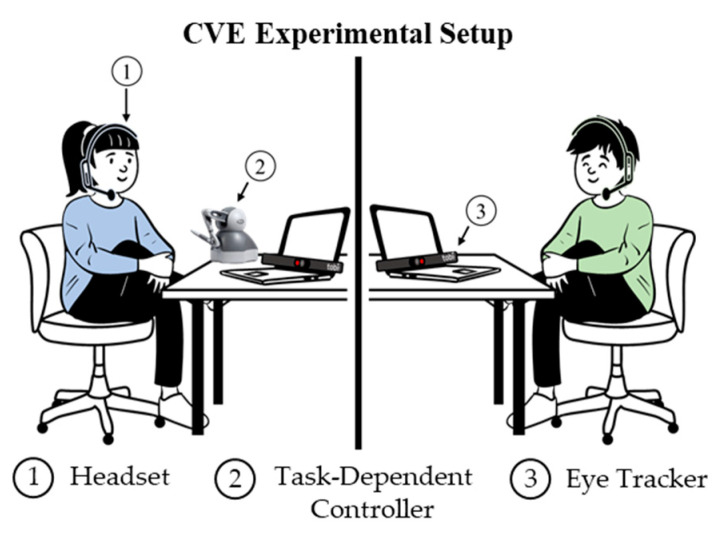
Experimental setup for data collection.

**Figure 3 sensors-23-03524-f003:**
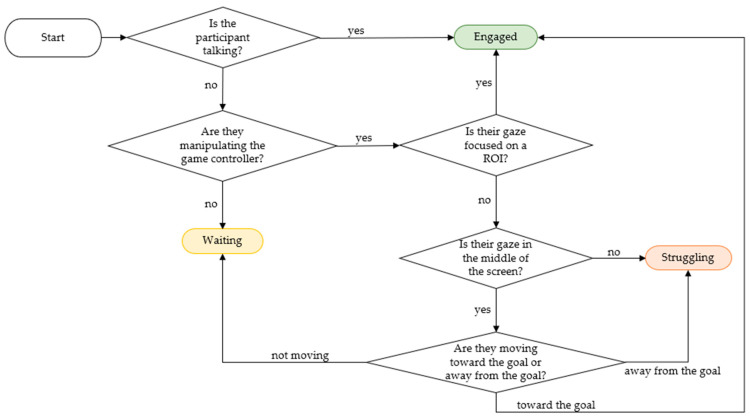
Flow chart for manual data annotation.

**Figure 4 sensors-23-03524-f004:**
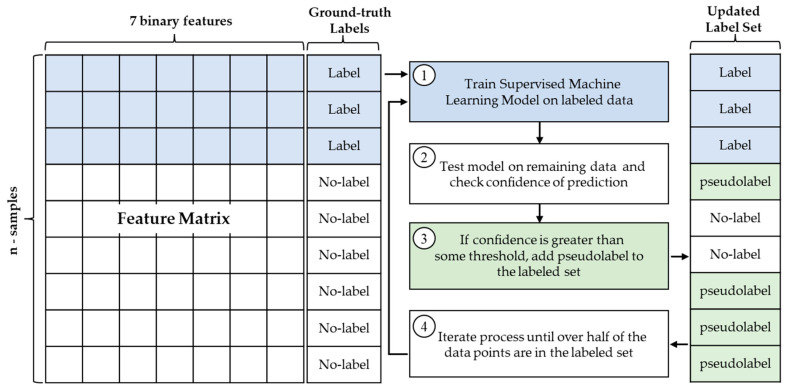
Semi-supervised self-training algorithm.

**Figure 5 sensors-23-03524-f005:**
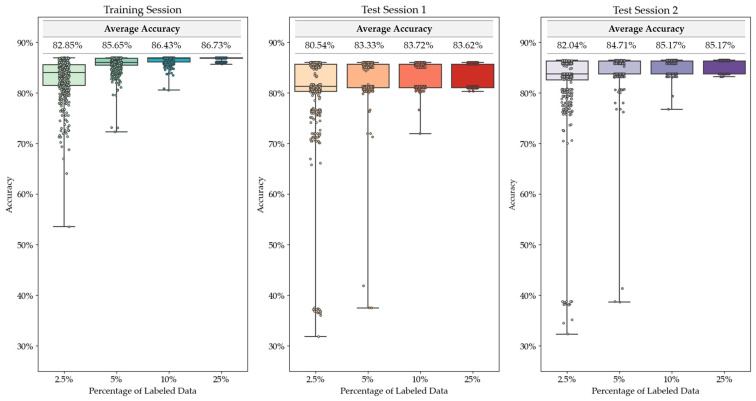
Semi-supervised self-training results across 500 trials.

**Figure 6 sensors-23-03524-f006:**
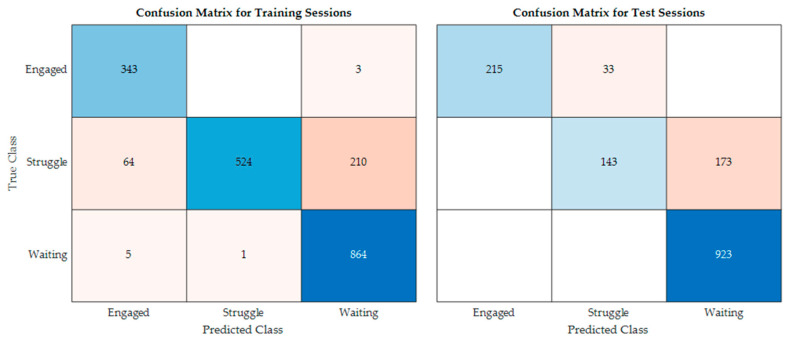
Confusion matrices.

**Figure 7 sensors-23-03524-f007:**
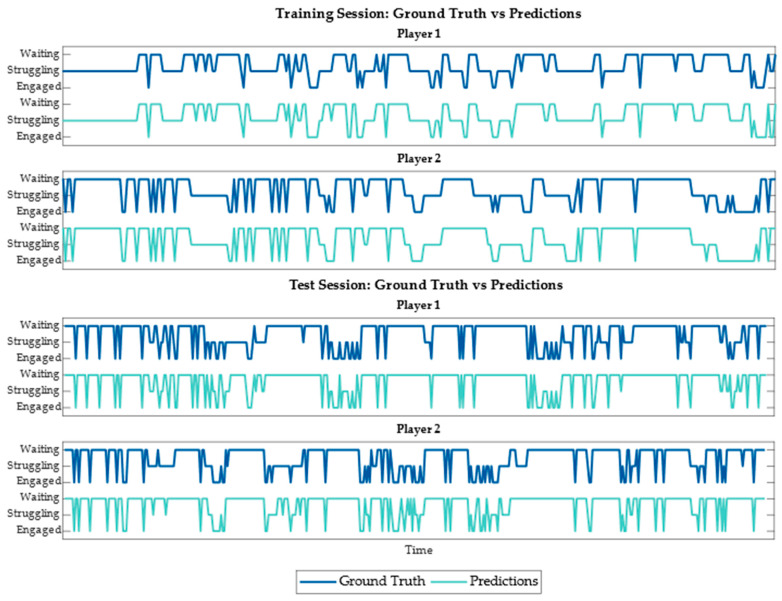
State progression charts.

**Figure 8 sensors-23-03524-f008:**
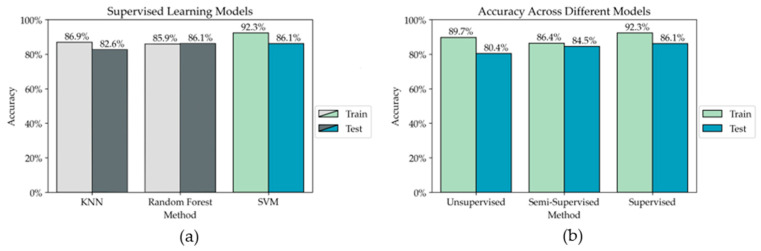
(**a**) Results of three supervised learning model. The best model is shown in green and blue. (**b**) Results using unsupervised learning, semi-supervised learning, and supervised learning.

**Table 1 sensors-23-03524-t001:** Description of the collaborative tasks.

Task	Input Method (Ease of Use)	Description	Aspects of Collaboration
Fulfillment Center	Mouse and Keyboard (Easy)	Drive a forklift to pick up and deliver crates from a storage shelf to a collection area in a warehouse	The two forklifts had varying height capacities making teamwork integral for success.
Furniture Assembly	Haptic Device (Required Extra Practice)	Assemble various pieces of furniture such as a coffee table and a bookcase using either assembly instructions or an image of the completed furniture.	The participants had varying information in their instructions. Also, each participant was tasked with different aspects of assembly making collaboration vital for assembly.
PC Assembly	Gamepad (Required Minimal Practice)	Build a computer by putting together different pieces of computer hardware.	The two users had different pieces of computer hardware as well as extra components. In addition, participants had mismatched assembly instructions making teamwork and communication essential for task completion.

**Table 2 sensors-23-03524-t002:** Participant metrics.

Metrics	ASD (N = 6)	TD (N = 6)
Age Mean (SD)	20.5 (2.81)	22.83 (3.60)
Gender (% Male)	50%	50%

**Table 3 sensors-23-03524-t003:** Features extracted from multimodal data with descriptions and examples.

Modality	Binary Feature	Description	Example of ‘1’ for Each Feature
Headset	Speech	If the *participant is speaking*, the feature is ‘1’ otherwise it is ‘0’.	Participant 1 is encouraging participant 2 in the PC assembly task by saying “Great Job!”
Task-Dependent Controller	Controller Activated	If the *controller is activated*, the feature is ‘1’ otherwise it is ‘0’.	Participant 1 moves the haptic controller in the furniture assembly task.
	Object Manipulated	If the *controller is activated in an area of interest*, the feature is ‘1’ otherwise it is ‘0’.	Participant 2 moves the haptic controller to move a table leg in the furniture assembly task.
	Moving Towards Goal	Uses distance to determine if the participant is *progressing toward* the goal. If they are, the feature is ‘1’ otherwise it is ‘0’.	Participant 1 is moving a table leg towards the desired position in the furniture assembly task.
	Moving Away from Goal	Uses distance to determine if the participant is *moving away from* to goal. If they are, the feature is ‘1’ otherwise it is ‘0’.	Participant 2 is struggling to move a table leg in the furniture assembly task and is moving away from the desired position.
Eye Tracker	Focused on Object	If the gaze is *focused on an area of interest*, the feature is ‘1’ otherwise it is ‘0’.	Participant 1 is looking at the motherboard in the PC assembly task.
	Not Focused on Screen	If the gaze *is not focused on the middle of the screen*, the feature is ‘1’ otherwise it is ‘0’.	Participant 2 is looking at the couch in the furniture assembly task which is in the outer portion of the screen.

## Data Availability

The data presented in this study are available on request from the corresponding author. The data are not publicly available due to the privacy of our participants and the requirements of our IRB.

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
