# Peer review of "Semi-Supervised Behavior Labeling Using Multimodal Data during Virtual Teamwork-Based Collaborative Activities"

_sensors, 2023, doi:10.3390/s23073524_

Round 1

Reviewer 1 Report

There are many concerns with the article such as:

Too many long lines make it difficult to understand the concept such as: “

1.       Adaptive human-computer systems require the recognition of human behavior states to 10 provide real-time feedback to scaffold skill learning and are being researched extensively for inter-11 vention and training in individuals with autism spectrum disorder (ASD).”. etc.

2.      Add suitable mathematics related to the proposed model.

3.      Add more competitive literature with suitable gaps in the literature.

4.      Add more detail so of the Flow Chart for Manual Data Annotation.

5.      Define how authors have optimized the hyper parameters of the proposed model. If it is on trail and error basis you should add as future scope. Consider “Evolving fusion-based visibility restoration model for hazy remote sensing images using dynamic differential evolution”

6.      Add more comparative results by considering competitive approaches. Add suitable discussion of the proposed model.

Author Response

Response to Reviewer 1 Comments

Point 1: Too many long lines make it difficult to understand the concept such as: “Adaptive human-computer systems require the recognition of human behavior states to provide real-time feedback to scaffold skill learning and are being researched extensively for intervention and training in individuals with autism spectrum disorder (ASD).”. etc.

Response 1: Thank you for pointing out the long sentences. We agree these need to be addressed to make the manuscript more understandable. In the specific example cited, we split it into two sentences to address your concern. In addition, we thoroughly reviewed the remainder of the manuscript to ensure that all other long sentences are broken up for ease of understanding.

Point 2: Add suitable mathematics related to the proposed model.

Response 2: We thank the reviewer for pointing out the omission of mathematics related to the model. To address this concern, we included the sigmoid equation used to calculate the predictions' posterior probability, which is the same as the confidence of the predictions. In addition, we discussed the SVM model used in more detail.

Point 3: Add more competitive literature with suitable gaps in the literature.

Response 3: We thank the reviewer for highlighting omissions from our literature survey. To address this comment, additional literature has been included. For example, we added two articles (one was a 31-article review) on the use of VR to train various life skills. Additionally, more literature relevant to machine learning (specifically semi-supervised learning and SVM) was added to further explain our chosen methodology.

Point 4: Add more detail so of the Flow Chart for Manual Data Annotation.

Response 4: We thank the reviewer for this comment. However, the flow chart shown is the flow chart that was used for Manual Data Annotation therefore there are no additional details we can add.

Point 5: Define how authors have optimized the hyper parameters of the proposed model. If it is on trail and error basis you should add as future scope. Consider “Evolving fusion-based visibility restoration model for hazy remote sensing images using dynamic differential evolution”

Response 5: We thank the reviewer for pointing out the omission of hyperparameter tuning. We have added more details on how this was completed. We utilized MATLABs built-in function fitcsvm which automatically optimizes the hyperparameters when training the model. Therefore, the hyperparameters are tuned at each iteration of the semi-supervised model. We added these details to the manuscript to address your concerns.

Point 6: Add more comparative results by considering competitive approaches. Add suitable discussion of the proposed model.

Response 6: Thank you for calling to our attention the lack of comparative results. To address this, we added two more supervised models for comparison. They are shown in Figure 8a.

Reviewer 2 Report

Article seems carrying the emerging topic for the future/current researchers. However, following minor revisions can be taken into account, please:

Key contributions shall be added in bullet form.

Limitations if the work shall be added.

Many Thanks

Reviewer 3 Report

This paper provides a concise overview of the importance of adaptive HCI systems for individuals with ASD and how virtual reality systems can be used to train teamwork skills. The authors also propose a semi-supervised machine learning method to supplement manual data labeling in CVE.

However, there are some areas where the authors could improve. For instance, it would be helpful to provide more context on the current state of research in adaptive HCI systems and how the proposed method fits into the existing literature. It is recommended to add literature in relevant AR/VR research contexts, such as: Prototyping an online virtual simulation course platform for college students to learn creative thinking. Systems. 2023; 11(2):89.

Additionally, the authors could benefit from more information on the specific techniques and algorithms used in the proposed semi-supervised learning model. More details on the model's architecture, training process, and performance metrics would help readers understand the research approach better. It is recommended to add literature in relevant contexts, such as: A survey on semi-supervised learning. Machine learning, 109(2), 373-440.

The authors could provide more information on the potential impact of their research, particularly in terms of improving employment prospects for individuals with ASD. Describing how the proposed method could be applied in real-world settings and the benefits it could bring to individuals with ASD would help make the research more relevant and impactful.

Overall, this paper provides a clear overview of the research topic and the proposed method but could be improved with more context and details on the specific techniques used and the potential impact of the research.
